# GEOM-GCN: GEOMETRIC GRAPH CONVOLUTIONAL NETWORKS

**Hongbin Pei**[1,5,*]
peihb15@mails.jlu.edu.cn

**Bingzhe Wei**[2]
bwei6@illinois.edu

**Kevin Chen-Chuan Chang**[2,3]
kcchang@illinois.edu

**Yu Lei**[4]
csylei@comp.polyu.edu.hk

**Bo Yang**[1,5,†]
ybo@jlu.edu.cn

[1]College of Computer Science and Technology, Jilin University, China
[2]Department of Electrical and Computer Engineering, University of Illinois at Urbana-Champaign, USA
[3]Department of Computer Science, University of Illinois at Urbana-Champaign, USA
[4]Department of Computing, Hong Kong Polytechnic University, Hong Kong
[5]Key Laboratory of Symbolic Computation and Knowledge Engineering of Ministry of Education, China

## ABSTRACT

Message-passing neural networks (MPNNs) have been successfully applied to representation learning on graphs in a variety of real-world applications. However, two fundamental weaknesses of MPNNs' aggregators limit their ability to represent graph-structured data: losing the structural information of nodes in neighborhoods and lacking the ability to capture long-range dependencies in disassortative graphs. Few studies have noticed the weaknesses from different perspectives. From the observations on classical neural network and network geometry, we propose a novel geometric aggregation scheme for graph neural networks to overcome the two weaknesses. The behind basic idea is the aggregation on a graph can benefit from a continuous space underlying the graph. The proposed aggregation scheme is permutation-invariant and consists of three modules, node embedding, structural neighborhood, and bi-level aggregation. We also present an implementation of the scheme in graph convolutional networks, termed Geom-GCN, to perform transductive learning on graphs. Experimental results show the proposed Geom-GCN achieved state-of-the-art performance on a wide range of open datasets of graphs.

## 1 INTRODUCTION

Message-passing neural networks (MPNNs), such as GNN (Scarselli et al., 2008), ChebNet (Defferrard et al., 2016), GG-NN (Li et al., 2016), GCN (Kipf & Welling, 2017), are powerful for learning on graphs with various applications ranging from brain networks to online social network (Gilmer et al., 2017; Wang et al., 2019). In a layer of MPNNs, each node sends its feature representation, a "message", to the nodes in its neighborhood; and then updates its feature representation by aggregating all "messages" received from the neighborhood. The neighborhood is often defined as the set of adjacent nodes in graph. By adopting permutation-invariant aggregation functions (e.g., summation, maximum, and mean), MPNNs are able to learn representations which are invariant to isomorphic graphs, i.e., graphs that are topologically identical.

Although existing MPNNs have been successfully applied in a wide variety of scenarios, two fundamental weaknesses of MPNNs' aggregators limit their ability to represent graph-structured data. Firstly, *the aggregators lose the structural information of nodes in neighborhoods*. Permutation invariance is an essential requirement for any graph learning method. To meet it, existing MPNNs adopt permutation-invariant aggregation functions which treat all "messages" from neighborhood as

---

*This work is conducted partially during his visit at University of Illinois at Urbana-Champaign.
†Corresponding author.

a set. For instance, GCN simply sums the normalized "messages" from all one-hop neighbors (Kipf & Welling, 2017). Such aggregation loses the structural information of nodes in neighborhood because it does not distinguish the "messages" from different nodes. Therefore, after such aggregation, we cannot know which node contributes what to the final aggregated output.

Without modeling such structural information, as shown in (Kondor et al., 2018) and (Xu et al., 2019), the existing MPNNs cannot discriminate between certain non-isomorphic graphs. In those cases, MPNNs may map non-isomorphic graphs to the same feature representations, which is obviously not desirable for graph representation learning. Unlike MPNNs, classical convolutional neural networks (CNNs) avoid this problem by using aggregators (i.e., convolutional filters) with a structural receiving filed defined on grids, i.e., a Euclidean space, and are hence able to distinguish each input unit. As shown by our experiments, such structural information often contains clues regarding topology patterns in graph (e.g., hierarchy), and should be extracted and used to learn more discriminating representations for graph-structured data.

Secondly, *the aggregators lack the ability to capture long-range dependencies in disassortative graphs.* In MPNNs, the neighborhood is defined as the set of all neighbors one hop away (e.g., GCN), or all neighbors up to $r$ hops away (e.g., ChebNet). In other words, only messages from nearby nodes are aggregated. The MPNNs with such aggregation are inclined to learn similar representations for proximal nodes in a graph. This implies that they are probably desirable methods for assortative graphs (e.g., citation networks (Kipf & Welling, 2017) and community networks (Chen et al., 2019)) where node homophily holds (i.e., similar nodes are more likely to be proximal, and vice versa), but may be inappropriate to the disassortative graphs (Newman, 2002) where node homophily does not hold. For example, Ribeiro et al. (2017) shows disassortative graphs where nodes of the same class exhibit high structural similarity but are far apart from each other. In such cases, the representation ability of MPNNs may be limited significantly, since they cannot capture the important features from distant but informative nodes.

A straightforward strategy to address this limitation is to use a multi-layered architecture so as to receive "messages" from distant nodes. For instance, due to the localized nature of convolutional filters in classical CNNs, a single convolutional layer is similarly limited in its representational ability. CNNs typically use multiple layers connected in a hierarchical manner to learn complex and global representations. However, unlike CNNs, it is difficult for multi-layer MPNNs to learn good representations for disassortative graphs because of two reasons. On one hand, relevant messages from distant nodes are mixed indistinguishably with a large number of irrelevant messages from proximal nodes in multi-layer MPNNs, which implies that the relevant information will be "washed out" and cannot be extracted effectively. On the other hand, the representations of different nodes would become very similar in multi-layer MPNNs, and every node's representation actually carries the information about the entire graph (Xu et al., 2018).

In this paper, we overcome the aforementioned weaknesses of graph neural networks starting from two basic observations: i) Classical neural networks effectively address the similar limitations thanks to the stationarity, locality, and compositionality in a continuous space (Bronstein et al., 2017); ii) The notion of network geometry bridges the gap between continuous space and graph (Hoff et al., 2002; Muscoloni et al., 2017). Network geometry aims to understand networks by revealing the latent continuous space underlying them, which assumes that nodes are sampled discretely from a latent continuous space and edges are established according to their distance. In the latent space, complicated topology patterns in graphs can be preserved and presented as intuitive geometry, such as subgraph (Narayanan et al., 2016), community (Ni et al., 2019), and hierarchy (Nickel & Kiela, 2017; 2018). Inspired by those two observations, we raise an enlightening question about the aggregation scheme in graph neural network.

- Can the aggregation on a graph benefit from a continuous latent space, such as using geometry in the space to build structural neighborhoods and capture long-range dependencies in the graph?

To answer the above question, we propose a novel aggregation scheme for graph neural networks, termed the *geometric aggregation scheme*. In the scheme, we map a graph to a continuous latent space via node embedding, and then use the geometric relationships defined in the latent space to build structural neighborhoods for aggregation. Also, we design a bi-level aggregator operating on the structural neighborhoods to update the feature representations of nodes in graph neural networks, which are able to guarantee permutation invariance for graph-structured data. Compared with exist-

ing MPNNs, the scheme extracts more structural information of the graph and can aggregate feature representations from distant nodes via mapping them to neighborhoods defined in the latent space.

We then present an implementation of the geometric aggregation scheme in graph convolutional networks, which we call *Geom-GCN*, to perform transductive learning, node classification, on graphs. We design particular geometric relationships to build the structural neighborhood in Euclidean and hyperbolic embedding space respectively. We choose different embedding methods to map the graph to a suitable latent space for different applications, where suitable topology patterns of graph are preserved. Finally, we empirically validate and analyze Geom-GCN on a wide range of open datasets of graphs, and Geom-GCN achieved the state-of-the-art results.

In summary, the contribution of this paper is three-fold: i) We propose a novel geometric aggregation scheme for graph neural network, which operates in both graph and latent space, to overcome the aforementioned two weaknesses; ii) We present an implementation of the scheme, Geom-GCN, for transductive learning in graph; iii) We validate and analyze Geom-GCN via extensive comparisons with state-of-the-art methods on several challenging benchmarks.

## 2 GEOMETRIC AGGREGATION SCHEME

In this section, we start by presenting the geometric aggregation scheme, and then outline its advantages and limitations compared to existing works. As shown in Fig. 1, the aggregation scheme consists of three modules, node embedding (panel A1 and A2), structural neighborhood (panel B1 and B2), and bi-level aggregation (panel C). We will elaborate on them in the following.

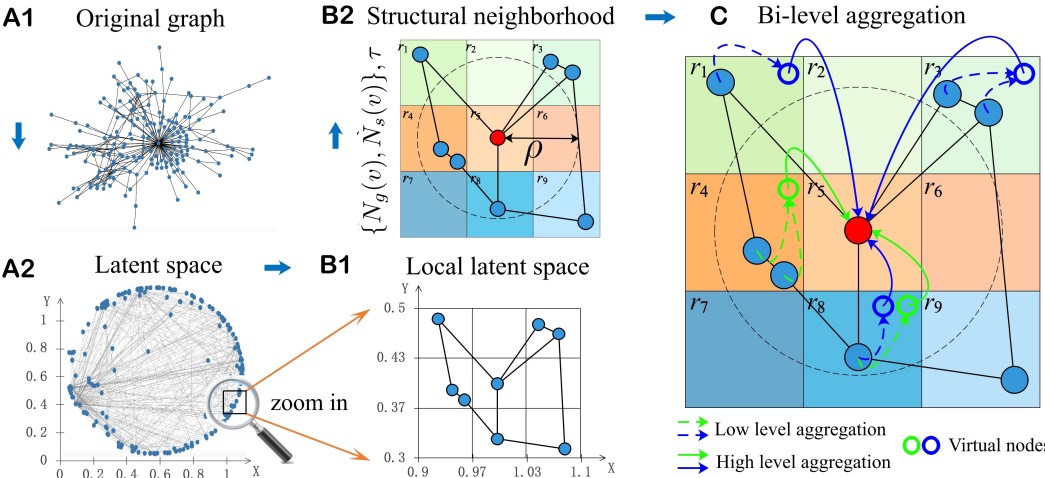

Figure 1: An illustration of the geometric aggregation scheme. **A1-A2** The original graph is mapped to a latent continuous space. **B1-B2** The structural neighborhood. All adjacent nodes lie in a small region around a center node in B1 for visualization. In B2, the neighborhood in the graph contains all adjacent nodes in graph; the neighborhood in the latent space contains the nodes within the dashed circle whose radius is $\rho$. The relational operator $\tau$ is illustrated by a colorful $3 \times 3$ grid where each unit is corresponding to a geometric relationship to the red target node. **C** Bi-level aggregation on the structural neighborhood. Dashed and solid arrows denote the low-level and high-level aggregation, respectively. Blue and green arrows denote the aggregation on the neighborhood in the graph and the latent space, respectively.

**A. Node embedding**. This is a fundamental module which maps the nodes in a graph to a latent continuous space. Let $\mathcal{G} = (V, E)$ be a graph, where each node $v \in V$ has a feature vector $\boldsymbol{x}_v$ and each edge $e \in E$ connects two nodes. Let $f : v \to \boldsymbol{z}_v$ be a mapping function from a node in graph to a representation vector. Here, $\boldsymbol{z}_v \in \mathbb{R}^d$ can also be considered as the position of node $v$ in a latent continuous space, and $d$ is the number of dimensions of the space. During the mapping, the structure and properties of graph are preserved and presented as the geometry in the latent space. For instance, hierarchical pattern in graph is presented as the distance to the original in embedding hyperbolic space (Nickel & Kiela, 2017). One can employ various embedding methods to infer the latent space (Cai et al., 2018; Wang et al., 2018).

**B. Structural neighborhood**. Based on the graph and the latent space, we then build a structural neighborhood, $\mathcal{N}(v) = (\{N_g(v), N_s(v)\}, \tau)$, for the next aggregation. The structural neighborhood consists of a set of neighborhood $\{N_g(v), N_s(v)\}$, and a relational operator on neighborhoods $\tau$.

The neighborhood in the graph, $N_g(v) = \{u | u \in V, (u, v) \in E\}$, is the set of adjacent nodes of $v$. The neighborhood in the latent space, $N_s(v) = \{u | u \in V, d(\boldsymbol{z}_u, \boldsymbol{z}_v) < \rho\}$, is the set of nodes from which the distance to $v$ is less than a pre-given parameter $\rho$. The distance function $d(\cdot, \cdot)$ depends on the particular metric in the space. Compared with $N_g(v)$, $N_s(v)$ may contain nodes which are far from $v$ in the graph, but have a certain similarity with $v$, and hence are mapped together with $v$ in the latent space though preserving the similarity. By aggregating on such neighborhood $N_s(v)$, the long-range dependencies in disassortative graphs can be captured.

The relational operator $\tau$ is a function defined in the latent space. It inputs an ordered position pair $(\boldsymbol{z}_v, \boldsymbol{z}_u)$ of nodes $v$ and $u$, and outputs a discrete variable $r$ which indicates the geometric relationship from $v$ to $u$ in the latent space. For $u, v \in V$,

$$\tau : (\boldsymbol{z}_v, \boldsymbol{z}_u) \to r \in R,$$

where $R$ is the set of the geometric relationships. According to the particular latent space and application, $r$ can be specified as an arbitrary geometric relationship of interest. A requirement on $\tau$ is that it should guarantee that each ordered position pair has only one geometric relationship. For example, $\tau$ is illustrated in Fig. 1B by a colorful $3 \times 3$ grid in a 2-dimensional Euclidean space, in which each unit is corresponding to a geometric relationship to node $v$.

**C. Bi-level aggregation**. With the structural neighborhood $\mathcal{N}(v)$, we propose a novel bi-level aggregation scheme for graph neural network to update the hidden features of nodes. The bi-level aggregation consists of two aggregation functions and operates in a neural network layer. It can extract effectively structural information of nodes in neighborhoods as well as guarantee permutation invariance for graph. Let $\boldsymbol{h}_v^l$ be the hidden features of node $v$ at the $l$-th layer, and $\boldsymbol{h}_v^0 = \boldsymbol{x}_v$ be the node features. The $l$-th layer updates $\boldsymbol{h}_v^l$ for every $v \in V$ by the following.

$$
\begin{aligned}
\boldsymbol{e}_{(i,r)}^{v,l+1} &= p(\{\boldsymbol{h}_u^l | u \in N_i(v), \tau(\boldsymbol{z}_v, \boldsymbol{z}_u) = r\}), \forall i \in \{g, s\}, \forall r \in R & \text{(Low-level aggregation)} \\
\boldsymbol{m}_v^{l+1} &= \underset{i \in \{g,s\}, r \in R}{q} ((\boldsymbol{e}_{(i,r)}^{v,l+1}, (i, r))) & \text{(High-level aggregation)} \\
\boldsymbol{h}_v^{l+1} &= \sigma(\boldsymbol{W}_l \cdot \boldsymbol{m}_v^{l+1}) & \text{(Non-linear transform)}
\end{aligned}
\tag{1}
$$

In the low-level, the hidden features of nodes that are in the same neighborhood $i$ and have the same geometric relationship $r$ are aggregated to a virtual node via the aggregation function $p$. The features of the virtual node are $\boldsymbol{e}_{(i,r)}^{v,l+1}$, and the virtual node is indexed by $(i, r)$ which is corresponding to the combination of a neighborhood $i$ and a relationship $r$. It is required to adopt a permutation-invariant function for $p$, such as an $L_p$-norm (the choice of $p = 1, 2,$ or $\infty$ results in average, energy, or max pooling). The low level aggregation is illustrated by dashed arrows in Fig. 1C.

In the high-level, the features of virtual nodes are further aggregated by function $q$. The inputs of function $q$ contain both the features of virtual nodes $\boldsymbol{e}_{(i,r)}^{v,l+1}$ and the identity of virtual nodes $(i, r)$. That is, $q$ can be a function that take an ordered object as input, e.g., concatenation, to distinguish the features of different virtual nodes, thereby extracting the structural information in the neighborhoods explicitly. The output of high-level aggregation is a vector $\boldsymbol{m}_v^{l+1}$. Then new hidden features of $v$, $\boldsymbol{h}_v^{(l+1)}$, are given by a non-linear transform, wherein $\boldsymbol{W}_l$ is a learnable weight matrix on the $l$-th layer shared by all nodes, and $\sigma(\cdot)$ is a non-linear activation function, e.g., a ReLU.

Permutation invariance is an essential requirement for aggregators in graph neural networks. Thus, we then prove that the proposed bi-level aggregation, Eq. 1, is able to guarantee invariance for any permutation of nodes. We firstly give a definition for permutation-invariant mapping of graph.

**Definition 1.** *Let a bijective function $\psi : V \to V$ be a permutation for nodes, which renames $v \in V$ as $\psi(v) \in V$. Let $V'$ and $E'$ be the node and edge set after a permutation $\psi$, respectively. A mapping of graph, $\phi(\mathcal{G})$, is permutation-invariant if, given any permutation $\psi$, we have $\phi(\mathcal{G}) = \phi(\mathcal{G}'), \mathcal{G}' = (V', E')$.*

**Lemma 1.** *For a composite function $\phi_1 \circ \phi_2(\mathcal{G})$, if $\phi_2(\mathcal{G})$ is permutation-invariant, the entire composite function $\phi_1 \circ \phi_2(\mathcal{G})$ is permutation-invariant.*

*Proof.* Let $\mathcal{G}'$ be an isomorphic graph of $\mathcal{G}$ after a permutation $\psi$, as defined in Definition 1. If $\phi_2(\mathcal{G})$ is permutation-invariant, we have $\phi_2(\mathcal{G}) = \phi_2(\mathcal{G}')$. Therefore, the entire composite function $\phi_1 \circ \phi_2(\mathcal{G})$ is permutation-invariant because $\phi_1 \circ \phi_2(\mathcal{G}) = \phi_1 \circ \phi_2(\mathcal{G}')$. □

**Theorem 1.** *Given a graph $\mathcal{G} = (V, E)$ and its structural neighborhood $\mathcal{N}(v), \forall v \in V$, the bi-level aggregation, Eq. 1, is a permutation-invariant mapping of graph.*

*Proof.* The bi-level aggregation, Eq. 1, is a composite function, where the low-level aggregation is the input of the high-level aggregation. Thus, Eq. 1 is permutation-invariant if the low-level aggregation is permutation-invariant according to Lemma 1.

We then prove that the low-level aggregation is permutation-invariant. The low-level aggregation consists of $2 \times |R|$ sub-aggregations, each of which is corresponding to the nodes in a neighborhood $i$ and with a relationship $r$ to $v$. Firstly, the input of each sub-aggregations is permutation-invariant because both $i \in \{g, s\}$ and $r \in R$ are determined by the given structural neighborhood $\mathcal{N}(v), \forall v \in V$, which is constant for any permutation. Secondly, Eq. 1 adopts a permutation-invariant aggregation function $p$ for the sub-aggregations. Thus the low-level aggregation is permutation-invariant. □

## 2.1 COMPARISONS TO RELATED WORK

We now discuss how the proposed geometric aggregation scheme overcomes the two aforementioned weaknesses, i.e., how it effectively models the structural information and captures the long-range dependencies, in comparison to some closely related works.

To overcome the first weakness of MPNNs, i.e., losing the structural information of nodes in neighborhoods, the proposed scheme explicitly models the structural information by exploiting the geometric relationship between nodes in latent space and then extracting the information effectively by using the bi-level aggregations. In contrast, several existing works attempt to learn some implicit structure-like information to distinguish different neighbors when aggregating features. For example, GAT (Velickovic et al., 2017), LGCL (Gao et al., 2018) and GG-NN (Li et al., 2016) learn weights on "messages" from different neighbors by using attention mechanisms and node and/or edge attributes. CCN (Kondor et al., 2018) utilizes a covariance architecture to learn structure-aware representations. The major difference between these works and ours is that we offer an explicit and interpretable way to model the structural information of nodes in neighborhood, with the assistance of the geometry in a latent space. We note that our work is orthogonal with existing methods and thus can be readily incorporated to further improve their performance. In particular, we exploit geometric relationships from the aspect of *graph topology*, while other methods focus on that of *feature representation*– the two aspects are complementary.

For the second weakness of MPNNs, i.e., lacking the ability to capture long-range dependencies, the proposed scheme models the long-range dependencies in disassortative graphs in two different ways. First of all, the distant (but similar) nodes in the graph can be mapped into a latent-space-based neighborhood of the target node, and then their useful feature representations can be used for aggregations. This way depends on an appropriate embedding method, which is able to preserve the similarities between the distant nodes and the target node. On the other hand, the structural information enables the method to distinguish different nodes in a graph-based neighborhood (as mentioned above). The informative nodes may have some special geometric relationships to the target node (e.g., a particular angle or distance), whose relevant features hence will be passed to the target node with much higher weights, compared to the uninformative nodes. As a result, the long-range dependencies are captured indirectly through the whole message propagation process in all graph-based neighborhoods. In literature, a recent method JK-Nets (Xu et al., 2018) captures the long-range dependencies by skipping connections during feature aggregations.

### 2.1.1 CASE STUDY ON DISTINGUISHING NON-ISOMORPHIC GRAPHS

In literature, Kondor et al. (2018) and Xu et al. (2019) construct several non-isomorphic example graphs that cannot be distinguished by the aggregators (e.g., mean and maximum) in existing MPNNs. We present a case study to illustrate how to distinguish the non-isomorphic example graphs once the structural neighborhood is applied. We take two non-isomorphic graphs in (Xu et al., 2019) as an example, where each node has the same feature $a$ and after any mapping $f(a)$ remains the same across all nodes, as shown in Fig. 2 (left). Then the aggregator, e.g., mean or maximum, over $f(a)$ remains $f(a)$, and hence the final representations of the nodes are the same. That is, mean and maximum aggregators fail to distinguish the two different graphs.

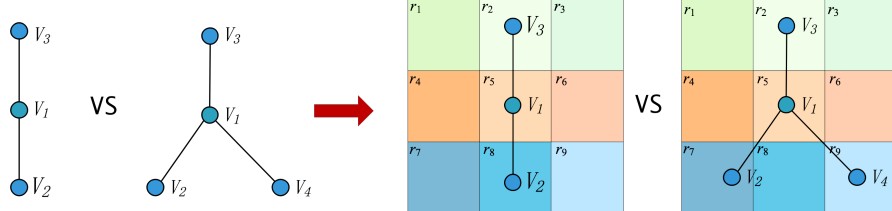

Figure 2: An illustration to distinguish non-isomorphic graphs by proposed structural neighborhood.

In contrast, the two graphs become distinguishable once we apply a structural neighborhood in aggregation. With the structural neighborhood, the nodes have different geometric relationships to the center node $V_1$ in the structural neighborhood, as shown in Fig. 2 (right). Taking aggregation for $V_1$ as an example, we can adopt different mapping function $f_r, r \in R$ to the neighbors with different geometric relationship $r$ to $V_1$. Then, the aggregator in two graph have different inputs, $\{f_2(a), f_8(a)\}$ in the left graph and $\{f_2(a), f_7(a), f_9(a)\}$ in the right graph. Finally, the aggregator (mean or maximum) will output different representations for the node $V_1$ in the two graphs, thereby distinguishing the topological difference between the two graphs.

## 3 GEOM-GCN: AN IMPLEMENTATION OF THE SCHEME

In this section, we present Geom-GCN, a specific implementation of the geometric aggregation scheme in graph convolutional networks, to perform transductive learning in graphs. To implement the general aggregation scheme, one needs to specify its three modules: node embedding, structural neighborhood, and bi-level aggregation function.

Node embedding is the fundamental. As shown in our experiments, a common embedding method which only preserves the connection and distance pattern in a graph can already benefit the aggregation. For particular applications, one can specify embedding methods to create suitable latent spaces where particular topology patterns (e.g., hierarchy) are preserved. We employ three embedding methods, Isomap (Tenenbaum et al., 2000), Poincare embedding (Nickel & Kiela, 2017), and struc2vec (Ribeiro et al., 2017), which result in three Geom-GCN variants: Geom-GCN-I, Geom-GCN-P, and Geom-GCN-S. Isomap is a widely used isometry embedding method, by which distance patterns (lengths of shortest paths) are preserved explicitly in the latent space. Poincare embedding and struc2vec can create particular latent spaces that preserve hierarchies and local structures in a graph, respectively. We use an embedding space of dimension 2 for ease of explanation.

The structural neighborhood $\mathcal{N}(v) = (\{N_g(v), N_s(v)\}, \tau)$ of node $v$ includes its neighborhoods in both the graph and latent space. The neighborhood-in-graph $N_g(v)$ consists of the set of $v$'s adjacent nodes in the graph, and the neighborhood-in-latent-space $N_s(v)$ those nodes whose distances to $v$ are less than a parameter $\rho$ in the latent space. We determine $\rho$ by increasing $\rho$ from zero until the average cardinality of $N_s(v)$ equals to that of $N_g(v)$, $\forall v \in V-$ i.e., when the average neighborhood sizes in the graph and latent spaces are the same. We use Euclidean distance in the Euclidean space. In the hyperbolic space, we approximate the geodesic distance between two nodes via their Euclidean distance in the local tangent plane.

Here we simply implement the geometric operator $\tau$ as four relationships of the relative positions between two nodes in a 2-D Euclidean or hyperbolic space. Particularly, the relationship set $R =$ {upper left, upper right, lower left, lower right}, and a $\tau(z_v, z_u)$ is given by Table 1. Note that, we adopt the rectangular coordinate system in the Euclidean space and angular coordinate in the hyperbolic space. By this way, the relationship "upper" indicates the node nearer to the origin and thus lie in a higher level in a hierarchical graph. One can design a more sophisticated operator $\tau$, such as borrowing the structure of descriptors in manifold geometry (Kokkinos et al., 2012; Monti et al., 2017), thereby preserving more and richer structural information in neighborhood.

Table 1: The relationship operator

| $\tau(z_v, z_u)$ | $z_v[0] > z_u[0]$ | $z_v[0] \leq z_u[0]$ |
|---|---|---|
| $z_v[1] \leq z_u[1]$ | upper left | upper right |
| $z_v[1] > z_u[1]$ | lower left | lower right |

Finally, to implement the bi-level aggregation, we adopt the same summation of normalized hidden features as GCN (Kipf & Welling, 2017) as the aggregation function $p$ in the low-level aggregation,

$$\boldsymbol{e}_{(i,r)}^{v,l+1} = \sum_{u \in N_i(v)} \delta(\tau(\boldsymbol{z}_v, \boldsymbol{z}_u), r)(\deg(v)\deg(u))^{\frac{1}{2}}\boldsymbol{h}_u^l, \ \ \forall i \in \{g,s\}, \forall r \in R,$$

where $\deg(v)$ is the degree of node $v$ in graph, and $\delta(\cdot,\cdot)$ is a Kronecker delta function that only allows the nodes with relationship $r$ to $v$ to be included. The features of all virtual nodes $\boldsymbol{e}_{(i,r)}^{v,l+1}$ are further aggregated in the high-level aggregation. The aggregation function $q$ is a concatenation $||$ for all layers except the final layer, which uses mean for its aggregation function. Then, the overall bi-level aggregation of Geom-GCN is given by

$$\boldsymbol{h}_v^{l+1} = \sigma(\boldsymbol{W}_l \cdot \mathop{||}_{i \in \{g,s\}} \mathop{||}_{r \in R} \boldsymbol{e}_{(i,r)}^{v,l+1})$$

where we use ReLU as the non-linear activation function $\sigma(\cdot)$ and $\boldsymbol{W}_l$ is the weight matrix to estimate by backpropagation.

## 4 EXPERIMENTS

We validate Geom-GCN by comparing Geom-GCN's performance with the performance of Graph Convolutional Networks (GCN) (Kipf & Welling (2017)) and Graph Attention Networks (GAT) (Velickovic et al. (2017)). Two state-of-the-art graph neural networks, on transductive node-label classification tasks on a wide variety of open graph datasets.

### 4.1 DATASETS

We utilize nine open graph datasets to validate the proposed Geom-GCN. An overview summary of characteristics of the datasets is given in Table 2.

Table 2: Datasets statistics

| Dataset | Cora | Cite. | Pubm. | Cham. | Squi. | Actor | Corn. | Texa. | Wisc. |
|---------|------|-------|-------|-------|-------|-------|-------|-------|-------|
| # Nodes | 2708 | 3327 | 19717 | 2277 | 5201 | 7600 | 183 | 183 | 251 |
| # Edges | 5429 | 4732 | 44338 | 36101 | 217073 | 33544 | 295 | 309 | 499 |
| # Features | 1433 | 3703 | 500 | 2325 | 2089 | 931 | 1703 | 1703 | 1703 |
| # Classes | 7 | 6 | 3 | 5 | 5 | 5 | 5 | 5 | 5 |

*Citation networks.* Cora, Citeseer, and Pubmed are standard citation network benchmark datasets (Sen et al., 2008; Namata et al., 2012). In these networks, nodes represent papers, and edges denote citations of one paper by another. Node features are the bag-of-words representation of papers, and node label is the academic topic of a paper.

*WebKB.* WebKB[1] is a webpage dataset collected from computer science departments of various universities by Carnegie Mellon University. We use the three subdatasets of it, Cornell, Texas, and Wisconsin, where nodes represent web pages, and edges are hyperlinks between them. Node features are the bag-of-words representation of web pages. The web pages are manually classified into the five categories, student, project, course, staff, and faculty.

*Actor co-occurrence network.* This dataset is the actor-only induced subgraph of the film-director-actor-writer network (Tang et al., 2009). Each nodes correspond to an actor, and the edge between two nodes denotes co-occurrence on the same Wikipedia page. Node features correspond to some keywords in the Wikipedia pages. We classify the nodes into five categories in term of words of actor's Wikipedia.

*Wikipedia network.* Chameleon and squirrel are two page-page networks on specific topics in Wikipedia (Rozemberczki et al., 2019). In those datasets, nodes represent web pages and edges are mutual links between pages. And node features correspond to several informative nouns in the Wikipedia pages. We classify the nodes into five categories in term of the number of the average monthly traffic of the web page.

---

[1]http://www.cs.cmu.edu/afs/cs.cmu.edu/project/theo-11/www/wwkb

## 4.2 Experimental setup

As mentioned in Section 3, we construct three Geom-GCN variants by using three embedding methods, Isomap (Geom-GCN-I), Poincare (Geom-GCN-P), and struc2vec (Geom-GCN-S). We specify the dimension of embedding space as two, and use the relationship operator $\tau$ defined in Table 1, and apply mean and concatenation as the low- and high- level aggregation function, respectively.

With the structural neighborhood, we perform a hyper-parameter search for all models on validation set. For fairness, the size of search space for each method is the same. The searching hyper-parameters include number of hidden unit, initial learning rate, weight decay, and dropout. We fix the number of layer to 2 and use Adam optimizer (Kingma & Ba, 2014) for all models. We use ReLU as the activation function for Geom-GCN and GCN, and ELU for GAT.

The final hyper-parameter setting is dropout of $p = 0.5$, initial learning rate of $0.05$, patience of 100 epochs, weight decay of $5E$-6 (WebKB datasets) or $5E$-5 (the other all datasets). In GCN, the number of hidden unit is 16 (Cora), 16 (Citeseer), 64 (Pubmed), 32 (WebKB), 48 (Wikipedia), and 32 (Actor). In Geom-GCN, the number of hidden unit is 8 times as many as the number in GCN since Geom-GCN has 8 virtual nodes. For each attention head in GAT, the number of hidden unit is 8 (Citation networks), 32 (WebKB), 48 (Wikipedia), and 32 (Actor). GAT has 8 attention heads in layer one and 8 (Pubmed) or 1 (the all other datasets) attention heads in layer two.

For all graph datasets, we randomly split nodes of each class into $60\%$, $20\%$, and $20\%$ for training, validation and testing. With the hyper-parameter setting, we report the average performance of all models on the test sets over 10 random splits.

## 4.3 Results and analysis

Results are summarized in Table 3. The reported numbers denote the mean classification accuracy in percent. In general, Geom-GCN achieves state-of-the-art performance. The best performing method is highlighted. From the results, Isomap embedding (Geom-GCN-I) which only preserves the connection and distance pattern in graph can already benefit the aggregation. We can also specify an embedding method to create a suitable latent space for a particular application (e.g., disassortative graph or hierarchical graph), by doing which a significant performance improvement is achieved (e.g., Geom-GCN-P).

Table 3: Mean Classification Accuracy (Percent)

| Dataset | Cora | Cite. | Pubm. | Cham. | Squi. | Actor | Corn. | Texa. | Wisc. |
|---|---|---|---|---|---|---|---|---|---|
| GCN | 85.77 | 73.68 | 88.13 | 28.18 | 23.96 | 26.86 | 52.70 | 52.16 | 45.88 |
| GAT | **86.37** | 74.32 | 87.62 | 42.93 | 30.03 | 28.45 | 54.32 | 58.38 | 49.41 |
| Geom-GCN-I | 85.19 | **77.99** | **90.05** | 60.31 | 33.32 | 29.09 | 56.76 | 57.58 | 58.24 |
| Geom-GCN-P | 84.93 | 75.14 | 88.09 | **60.90** | **38.14** | **31.63** | **60.81** | **67.57** | **64.12** |
| Geom-GCN-S | 85.27 | 74.71 | 84.75 | 59.96 | 36.24 | 30.30 | 55.68 | 59.73 | 56.67 |

### 4.3.1 Ablation study on contributions from two neighborhoods

The proposed Geom-GCN aggregates "message" from two neighborhoods which are defined in graph and latent space respectively. In this section, we present an ablation study to evaluate the contribution from each neighborhood though constructing new Geom-GCN variants with only one neighborhood. For the variants with only neighborhood in graph, we use "g" as a suffix of their name (e.g., Geom-GCN-I-g), and use suffix "s" to denote the variants with only neighborhood in latent space (e.g., Geom-GCN-I-s). Here we set GCN as a baseline so that the contribution can be measured via the performance improvement comparing with GCN. The results are summarized in Table 4, where positive improvement is denoted by an up arrow $\uparrow$ and negative improvement by a down arrow $\downarrow$. The best performing method is highlighted.

We also design an index denoted by $\beta$ to measure the homophily in a graph,

$$\beta = \frac{1}{|V|} \sum_{v \in V} \frac{\text{Number of } v\text{'s neighbors who have the same label as } v}{\text{Number of } v\text{'s neighbors}}.$$

A large $\beta$ value implies that the homophily, in term of node label, is strong in a graph, i.e., similar nodes tend to connect together. From Table 4, one can see that assortative graphs (e.g., citation networks) have a much larger $\beta$ than disassortative graphs (e.g., WebKB networks).

Table 4 exhibits three interesting patterns: i) Neighborhoods in graph and latent space both benefit the aggregation in most cases; ii) Neighborhoods in latent space have larger contributions in disassortative graphs (with a small $\beta$) than assortative ones, which implies relevant information from disconnected nodes is captured effectively by the neighborhoods in latent space; iii) To our surprise, several variants with only one neighborhood (in Table 4) achieve better performances than the variants with two neighborhoods (in Tabel 3). We think the reason is that Geom-GCN with two neighborhoods aggregate more irrelevant "messages" than Geom-GCN with only one neighborhood, and the irrelevant "messages" adversely affect the performance. Thus, we believe an attention mechanism can alleviate this issue– which we will study as future work.

Table 4: Mean Classification Accuracy (Percent)

| Dataset $\beta$ | Cora 0.83 | Cite. 0.71 | Pumb. 0.79 | Cham. 0.25 | Squi. 0.22 | Actor 0.24 | Corn. 0.11 | Texa. 0.06 | Wisc. 0.16 |
|---|---|---|---|---|---|---|---|---|---|
| Geom-GCN-I-g | 86.26 ↑0.48 | **80.64** ↑6.96 | **90.72** ↑2.59 | **68.00** ↑39.82 | **46.01** ↑22.05 | 31.96 ↑4.04 | 65.40 ↑12.70 | 72.51 ↑21.35 | 68.23 ↑22.35 |
| Geom-GCN-I-s | 77.34 ↓8.34 | 72.22 ↓1.46 | 85.02 ↓3.11 | 61.64 ↑33.46 | 37.98 ↑14.02 | 30.59 ↑2.67 | 62.16 ↑9.46 | 60.54 ↑8.38 | 64.90 ↑19.01 |
| Geom-GCN-P-g | 86.30 ↑0.52 | 75.45 ↑1.76 | 88.40 ↑0.27 | 63.07 ↑34.89 | 38.41 ↑14.45 | 31.55 ↑3.63 | 64.05 ↑11.35 | 73.05 ↑21.89 | 69.41 ↑23.53 |
| Geom-GCN-P-s | 73.14 ↓12.63 | 71.65 ↓2.04 | 86.95 ↓1.18 | 43.20 ↑15.02 | 30.47 ↑6.51 | **34.59** ↑6.67 | **75.40** ↑22.70 | **73.51** ↑21.35 | **80.39** ↑34.51 |
| Geom-GCN-S-g | **87.00** ↑1.23 | 75.73 ↑2.04 | 88.44 ↑0.31 | 67.04 ↑38.86 | 44.92 ↑20.96 | 31.27 ↑3.35 | 67.02 ↑14.32 | 71.62 ↑19.46 | 69.41 ↑23.52 |
| Geom-GCN-S-s | 66.92 ↓18.85 | 66.03 ↓7.65 | 79.41 ↓8.72 | 49.21 ↑21.03 | 31.27 ↑7.31 | 30.32 ↑2.40 | 62.43 ↑9.73 | 63.24 ↑11.08 | 64.51 ↑18.63 |

### 4.3.2 ANALYSIS OF EMBEDDING SPACE COMBINATION

The structural neighborhood in Geom-GCN is very flexible, where one can combine arbitrary embedding space. To study which combination of embedding spaces is desirable, we construct new Geom-GCN variants by adopting neighborhoods built by different embedding space. For the variants adopted Isomap and poincare embedding space to build neighborhood in graph and in latent space respectively, we use Geom-GCN-IP to denote it. The naming rule is the same for other combinations. The performances of all variants are summarized in Table 5. One can observe that several combinations achieve better performance than Geom-GCN with neighborhoods built by only one embedding space (in Table 3); and there are also many combinations that have bad performance. Thus, we think it's significant future work to design an end-to-end framework that can automatically determine the right embedding spaces for Geom-GCN.

Table 5: Mean Classification Accuracy (Percent)

| Dataset | Cora | Cite. | Pubm. | Cham. | Squi. | Actor | Corn. | Texa. | Wisc. |
|---|---|---|---|---|---|---|---|---|---|
| Geom-GCN-IP | 85.13 | **79.41** | **90.49** | 65.77 | **45.49** | **31.94** | **60.00** | 66.49 | 62.75 |
| Geom-GCN-PI | 85.09 | 75.08 | 85.64 | 59.19 | 32.65 | 29.16 | 58.11 | 58.11 | 58.63 |
| Geom-GCN-IS | 84.51 | 77.83 | 88.66 | 58.40 | 35.29 | 29.41 | 54.32 | 57.57 | 57.65 |
| Geom-GCN-SI | 85.31 | 75.50 | 85.52 | 62.13 | 32.57 | 28.97 | 57.30 | 60.00 | 55.10 |
| Geom-GCN-PS | **85.65** | 74.84 | 84.96 | 56.34 | 28.27 | 29.53 | 58.11 | 62.43 | 60.59 |
| Geom-GCN-SP | 85.43 | 75.71 | 88.00 | **65.81** | 44.53 | 31.16 | 58.38 | **67.84** | **65.10** |

### 4.3.3 ANALYSIS OF TIME COMPLEXITY

Time complexity is very important for graph neural networks because real-world graphs are always very large. In this subsection, we firstly present the theoretical time complexity of Geom-GCN and then compare the real running time of GCN, GAT, and Geom-GCN.

To update the representations of one node, the time complexity of Geom-GCN is $O(n \times m \times 2|R|)$ where $n$ is the size of input representations, $m$ is the number of hidden unit in non-linear transform

for each virtual node (i.e., $(i, r)$), and $2|R|$ is the number of virtual nodes. Geom-GCN has $2|R|$ times complexity than GCN whose time complexity is $O(n \times m)$.

We also compare the real running time (500 epochs) of GCN, GAT, and Geom-GCN on all datasets with the hyper-parameters described in Section 4.2. Results are shown in Fig. 3 (a). One can see that GCN is the fastest, and GAT and Geom-GCN are on the same level. An important future work is to develop accelerating technology so as to solve the scalability of Geom-GCN.

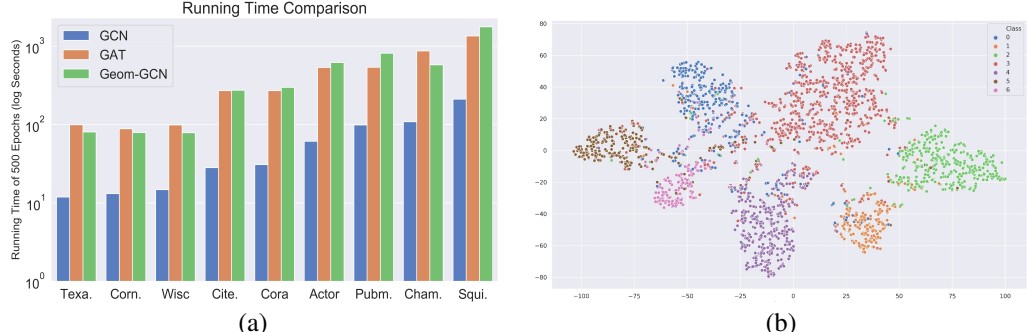

Figure 3: (a) Running time comparison. GCN, GAT, and Geom-GCN both run 500 epochs, and $y$ axis is the log seconds. GCN is the fastest, and GAT and Geom-GCN are on the same level. (b) A visualization for the feature representations of Cora obtained from Geom-GCN-P in a 2-D space. Node colors denote node labels. There are two obvious patterns, nodes with the same label exhibit a spatial clustering and all nodes distribute radially. The radial pattern indicates graph's hierarchy learned by Poincare embedding.

### 4.3.4 VISUALIZATION

To study what patterns are learned in the feature representations of node by Geom-GCN, we visualize the feature representations extracted by the last layer of Geom-GCN-P on Cora dataset by mapping it into a 2-D space though t-SNE (Maaten & Hinton, 2008), as shown in Fig. 3 (b). In the figure, the nodes with the same label exhibit spatial clustering, which could shows the discriminative power of Geom-GCN. That all nodes distribute radially in the figure indicates the proposed model learn graph's hierarchy by Poincare embedding.

### 4.4 CONCLUSION AND FUTURE WORK

We tackle the two major weaknesses of existing message-passing neural networks over graphs–losses of discriminative structures and long-range dependencies. As our key insight, we bridge a discrete graph to a continuous geometric space via graph embedding. That is, we exploit the principle of convolution: *spatial aggregation over a meaningful space*– and our approach thus extracts or "recovers" the lost information (discriminative structures and long-range dependencies) in an embedding space from a graph. We proposed a general geometric aggregation scheme and instantiated it with several specific Geom-GCN implementations, and our experiments validated clear advantages over the state-of-the-art. As future work, we will explore techniques for choosing a right embedding method– depending not only on input graphs but also on target applications, such as epidemic dynamic prediction on social contact network (Yang et al., 2017; Pei et al., 2018).

### ACKNOWLEDGMENTS

We thank the reviewers for their valuable feedback. This work was supported in part by National Natural Science Foundation of China under grant 61876069, 61572226 and 61902145, National Science Foundation IIS 16-19302 and IIS 16-33755, Jilin Province Key Scientific and Technological Research and Development project under grants 20180201067GX and 20180201044GX, University science and technology research plan project of Jilin Province under grants JJKH20190156KJ, Zhejiang University ZJU Research 083650, Futurewei Technologies HF2017060011 and 094013, UIUC OVCR CCIL Planning Grant 434S34, UIUC CSBS Small Grant 434C8U, Advanced Digital Sciences Center Faculty Grant, and China Scholarships Council under scholarship 201806170202. Any opinions, findings, and conclusions or recommendations expressed in this publication are those of the author(s) and do not necessarily reflect the views of the funding agencies.

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
