# OpenReview forum: "Geom-GCN: Geometric Graph Convolutional Networks"
_ICLR.cc/2020/Conference — Accept (Spotlight)_

### Official Review · AnonReviewer1 · 2019-10-22
**Official Blind Review #1**

**Rating:** 6

**Review:**


This work proposes geometric aggregation scheme for GCNs, which aims to overcome the limitations in traditional GCNs; those are lacking long distance dependencies and structure information in nodes. In particular, each node is transformed into a latent space. To overcome the first limitation, some nodes that are not directly connected but fall in a near range are also used in aggregation. A relational operator is used to provide position information for each pair of nodes. In this way, the structure information in graph can be used.

The method proposed in this work is novel and interesting. However, I am confused how this method can overcome the two limitations faced by previous GCNs. To my understanding, GEOM-GCN maps all node in to a 2D latent space. This can be treated as a lower-dimension representation for each node. Based on this, some similar nodes are clustered together. The relational operator is a kind of ranking operator that can rank two nodes based on latent space representations. If my understanding is wrong, please correct me.

Based on this understanding, I didn't find this method can solve the two limitations.

1. To overcome the long-term dependency limitation, GEOM-GCN selects some nodes that are close but not directly connected for aggregation. However, the selected nodes in this way may not connect to the center node. This is a issue that if two nodes that are not connected should be aggregated. The authors should clarify this.

2. The relational operator is used to provide a ranking between two nodes. However, how such kind of operators can be used to aggregate the structure information as described in GIN. For example, how to distinguish those example graphs using this work. I think it would be a plus if authors can make this clear in the paper.

3. The experimental studies are quite weak. Some ablation studies should be done to evaluate the contribution of each proposed methods. For example, how N_{s}(v) contributes to the performance. This is very important for fully evaluating your methods.

4. More tasks and datasets can be added such as graph classification and social networks.

5. Some notations are quite confusing. Like in eq.(1), why m is bold but W is not.

**Experience Assessment:**

I have published in this field for several years.

**Review Assessment: Checking Correctness Of Derivations And Theory:**

I assessed the sensibility of the derivations and theory.

**Review Assessment: Checking Correctness Of Experiments:**

I assessed the sensibility of the experiments.

**Review Assessment: Thoroughness In Paper Reading:**

I read the paper at least twice and used my best judgement in assessing the paper.

---

> ### Author Response · Authors · 2019-11-15
> **Reply to Official Review #1 Part 1**
>
> We thank the reviewer for the thoughtful comments. We have revised and updated the paper according to your suggestions and would like to answer questions as follows:
>
>
> Q1. To overcome the long-term dependency limitation, GEOM-GCN selects some nodes that are close but not directly connected for aggregation. However, the selected nodes in this way may not connect to the center node. This is an issue that if two nodes that are not connected should be aggregated. The authors should clarify this.
>
> Response:
>
> Long-term dependency is common in real-world graphs. For instance, in transcription networks, the sign-sensitive accelerators that speed up the response time of the target gene expression, are far apart from each other [Mangan 2003]. In the C. elegans frontal neuronal network, three-ring motor neurons that serve as the source of information, are also not directly connected [Kaiser 2006].
>
> However, the existing message passing neural networks (MPNNs), such as GCN, cannot handle such graphs because in those models relevant information from distant nodes is mixed with a large number of 1) proximal but irrelevant and 2) other distant (and truly irrelevant) nodes. This limitation has been analyzed deeply in JK-Net [Xu 2018].
>
> The issue of long-term dependency (from distant but relevant) can be addressed if the relevant information from distant nodes can be effectively filtered and aggregated in someway. For instance, TO-GCN could automatically connect two disconnected nodes in one class by topology optimization [Yang 2019].
>
> Thus, as the literature above and our own observations indicate, we believe long-term dependency is a true issue-- and to filter/aggregate relevant information from disconnected nodes is a solution we advocate, which this paper attempts to realize by mapping nodes to an embedded space where distant but relevant nodes become close.
>
> The key to this solution is how to identify those disconnected nodes with relevant information. In Geom-GCN we expect that such nodes can be mapped into the neighborhood of a center node in an embedded latent space. Then, the relevant information can be extracted via the aggregation in the latent space. We acknowledge that the efficacy of Geom-GCN would depend on the selected embedded space. From experiments, we indeed observe that relevant information is aggregated from disconnected nodes in disassortative graphs (see Table 4 in the revision). As future work, we will explore techniques for automatically choosing a right embedding method–- depending not only on input graphs but also on target applications.
>
>
> [Mangan 2003] Mangan, S., & Alon, U. Structure and function of the feed-forward loop network motif. Proceedings of the National Academy of Sciences (PNAS), 2003, 100(21), 11980-11985.
>
> [Kaiser 2006] Kaiser, M., & Hilgetag, C. C. Nonoptimal component placement, but short processing paths, due to long-distance projections in neural systems. PLoS computational biology, 2006, 2(7), e95.
>
> [Xu 2018] Xu, K., Li, C., Tian, Y., Sonobe, T., Kawarabayashi, K. I., & Jegelka, S. Representation learning on graphs with jumping knowledge networks. International Conference on Machine Learning (ICML). 2018, 5449–5458.
>
> [Yang 2019] Yang, L., Kang, Z., Cao, X., Jin, D., Yang, B., & Guo, Y. Topology optimization based graph convolutional network. In Proceedings of the Twenty-Eighth International Joint Conference on Artificial Intelligence, IJCAI. 2019, 4054-4061.
>
> Q2. The relational operator is used to provide a ranking between two nodes. However, how such kind of operators can be used to aggregate the structure information as described in GIN. For example, how to distinguish those example graphs using this work. I think it would be a plus if authors can make this clear in the paper.
>
> Response:
>
> Indeed, the non-isomorphic example graphs in GIN can be distinguished by applying simple aggregator, mean, or maximum, in a structural neighborhood. We provide and describe a detailed solution in the revision (see Section 2.1.1).

---

> > ### Author Response · Authors · 2019-11-15
> > **Reply to Official Review #1 Part 2**
> >
> > Q3. The experimental studies are quite weak. Some ablation studies should be done to evaluate the contribution of each proposed methods. For example, how N_{s}(v) contributes to the performance. This is very important for fully evaluating your methods.
> >
> > Response:
> >
> > Thanks for the suggestion. We totally agree and have added an ablation study in the revision to evaluate the contributions from graph neighborhood and latent space neighborhood, respectively.
> >
> > We indeed observed three interesting patterns in the experiment results (Table 4):
> > 1) Both graph and latent space neighborhoods benefit the aggregation in most cases;
> > 2) The latent space neighborhoods have larger contributions in disassortative graphs (with a small $\beta$) than assortative ones, which implies relevant information from disconnected nodes is aggregated effectively in the neighborhood of the latent space;
> > 3) To our surprise, several variants with only one neighborhood achieve better performances than the variants with both graph and latent space neighborhood. We think the reason is that Geom-GCN with both graph and latent neighborhoods aggregate more irrelevant "messages" than Geom-GCN with only one neighborhood, which adversely affects the performance. Thus, we believe an attention mechanism can alleviate this issue-- which we will study as future work.
> >
> > As time is limited in the rebuttal/discussion period, we only conduct this study on six small datasets, and a comprehensive study will be added in the future revision.
> >
> >
> > Q4. More tasks and datasets can be added such as graph classification and social networks.
> >
> > Response:
> >
> > Thanks for the suggestions. Graph classification is an important graph inductive learning task. The main challenge to extend the proposed Geom-GCN from current transductive learning (node classification) to inductive learning (graph classification) is that we need to align the embedded latent spaces of different graphs. Without space alignment, even two same graphs may result in different embedded latent spaces using the same embedding methods. In this situation, current Geom-GCN will fail to classify graphs. On the other hand, once the spaces are well aligned, we think Geom-GCN can be used to conduct graph inductive learning tasks. As the key issue is graph alignment in embedded space, which is orthogonal to our focus here-- and which appears to have limited existing work in the literature-- we believe it is interesting future work.
> >
> > For the social network dataset, we tried to run Geom-GCN on a large social network (Reddit data [Hamilton, 2017]), which has around 0.23 million nodes. However, we found that the Reddit graph is too large to be handled by current methods-- Geom-GCN, GCN, and GAT cannot handle this dataset. We will try our best to solve the scalability of Geom-GCN in the future.
> >
> > [Hamilton, 2017] Hamilton, W., Ying, Z., & Leskovec, J. Inductive representation learning on large graphs. In Advances in Neural Information Processing Systems. 2017, 1024-1034.
> >
> >
> > Q5. Some notations are quite confusing. Like in eq.(1), why m is bold but W is not.
> >
> > Response:
> >
> > We have corrected the formatting of W in the revision according to the standardized notation of ICLR 2020.
> >
> > We thank the reviewer once again. And we have released anonymously the code of Geom-GCN, and the particular architectures of Geom-GCN are also reported in the revision. One can access the code via link: https://github.com/anonymous-conference-submission/geom-gcn/.

---

### Official Review · AnonReviewer3 · 2019-10-27
**Official Blind Review #3**

**Rating:** 8

**Review:**

The work is based on the premise that existing MPNNs have two main weaknesses: (i) the loss function doesn't properly capture the spatial information during graph convolution, and (ii) the difficulty to manage the information encoded in the long range connections.

The main contribution of this work is a novel method called geometric aggregation. The proposed method is based on two elements: (i) a latent space mapping to capture spatial information using a new bi-level operator, (ii) an integration of geometric aggregation inside GCN, namely geom-GCN. More in detail the idea reported in this work is to map the input graph into an embedding where the geometric relations between nodes are preserved. The graph is embedded using a usual embedding function that guarantees to preserve some graph property of interest like the hierarchy of nodes. After the node embedding, the authors propose to create a structural neighbourhood both (i) in the latent space, taking the nodes within an arbitrary radius, and (ii) in the original space, by taking the adjacent nodes. The expectation here is that with the proper embedding the latent space can catch connections, which are long range in the original space. The message passing is actuated exploiting first the structural neighbourhood to do low-level aggregation, which aggregates nodes that have the same geometric relationship using permutation invariant operators, and then the result of these aggregations, which are virtual nodes, are aggregated again through high-level aggregation making use of operators like concatenation.

The goal of this work is clearly formulated by posing the proper research questions. The topic is relevant and it is part of the research agenda of ICLR. A key point of the proposed method is the ortogonalithy of geometric aggregation with respect to other aggregators like GAT. The design of the structural neighboorhood allows the network to choose which neighbors are the most important for the learning task.

Some minor comments.
The strong dependency from the embedding fuction does not guarantee the discovery of long range connections. It may happen that the proposed embedding does not catch the relevant information for the task; in these cases the a-priori knowledge on the task becomes crucial. This potential issue is partially supported by the results presented in the mauscript, where there is a gain only when the correct embedding is chosen.
A further critical point is the choice of the radius. Such a choice can be operated only with an empirical assessment. It is not clear whether it migth be meaningful to choose a radius thatwould encode the same neighbourhood as in the original sapce of data.
The authors claim that even if there are more hops between two nodes the relevant information would arrive from the far node to the target node. Nevertheless we may conceive a situation where the relevant information is washed out during the hops. It may happen when the information of the far node is relevant for the target node, but it is not relevant for the target neighbour nodes.
The use of concatenation as high level operator is critically dependent from the radius and from the number of edges in the graph. In cases of large values for radius or very dense graphs, the concatenation may increase the spatial complexity of the networks.
Concerning the Section on empirical analysis, it might be of interest to investigate whether with a proper number of layers a GCN would emulate a geom-GCN.

**Experience Assessment:**

I have read many papers in this area.

**Review Assessment: Checking Correctness Of Derivations And Theory:**

N/A

**Review Assessment: Checking Correctness Of Experiments:**

I did not assess the experiments.

**Review Assessment: Thoroughness In Paper Reading:**

I read the paper at least twice and used my best judgement in assessing the paper.

---

> ### Author Response · Authors · 2019-11-15
> **Reply to Official Review #3 Part 1**
>
> We would like to thank the reviewer for their thoughtful comments and appreciation. We would like to answer the reviewer’s questions as follows:
>
>
> Q1. The strong dependency from the embedding function does not guarantee the discovery of long range connections. It may happen that the proposed embedding does not catch the relevant information for the task; in these cases the a priori knowledge on the task becomes crucial. This potential issue is partially supported by the results presented in the manuscript, where there is a gain only when the correct embedding is chosen.
>
> Response:
>
> We acknowledge that the efficacy of Geom-GCN would depend on the selected embedded space. To further evaluate the influence on performance from the embedded space choice, we add two new sections in the revision (Section 4.3.1 and 4.3.2). The two sections analyze this issue from two different perspectives. Both of them indicate that some embedded spaces have a larger contribution/influence than the others. Thus, we believe it's significant future work to design an end-to-end framework that can automatically determine the right embedded space for Geom-GCN.
>
>
> Q2. A further critical point is the choice of the radius. Such a choice can be operated only with an empirical assessment. It is not clear whether it might be meaningful to choose a radius that would encode the same neighborhood as in the original space of data.
>
> Response:
>
> Thanks for your suggestion. Radius indeed is a very important hyper-parameters for Geom-GCN. When the radius is too small, relevant information cannot be aggregated comprehensively because the neighborhood in latent space becomes too small. And when the radius is too large, relevant information may be “washed out” by too much irrelevant information from the neighborhood in latent space. We will conduct sensibility tests for the radius on each dataset to determine a good radius in the future.
>
>
> Q3. The use of concatenation as high level operator is critically dependent from the radius and from the number of edges in the graph. In cases of large values for radius or very dense graphs, the concatenation may increase the spatial complexity of the networks.
>
> Response:
>
> Thanks for your suggestion. We agree that both the radius and the number of edges can increase the spatial complexity of Geom-GCN. However, what they affect is the low-level aggregation rather than the high-level aggregation. The reason is that the input of low-level aggregation is the representations of nodes in neighborhoods, and the input of high-level aggregation is the representations of virtual nodes, where the number of virtual nodes is fixed, i.e., 2|R|,|R| is the number of geometric relationships. Thus, we can employ concatenation for high-level aggregation when |R| is not too large.
>
>
> Q4. The authors claim that even if there are more hops between two nodes the relevant information would arrive from the far node to the target node. Nevertheless, we may conceive a situation where the relevant information is washed out during the hops. It may happen when the information of the far node is relevant for the target node, but it is not relevant for the target neighbor nodes.
>
> Response:
>
> A fundamental weakness of existing message passing neural networks (MPNNs) is lacking the ability to capture long-range dependencies. The reason is exactly what you mentioned, the relevant information is washed out during the many hops.
>
> This weakness can be addressed if the relevant information from distant nodes can be effectively filtered and aggregated in someway. For instance, TO-GCN could automatically connect two disconnected nodes in one class by topology optimization [Yang 2019]. In this paper, we attempt to map the distant nodes with relevant information into a small area in an embedded space. Then the relevant information can be aggregated effectively in the neighborhood defined in the embedded space.
>
> We acknowledge that the efficacy of such aggregation would depend on the selected embedded space. From experiments, we indeed observe that relevant information is aggregated from disconnected nodes in disassortative graphs (see Table 4 in the revision). As future work, we will explore techniques for automatically choosing a right embedding method–- depending not only on input graphs but also on target applications.
>
> [Yang 2019] Yang, L., Kang, Z., Cao, X., Jin, D., Yang, B., & Guo, Y. Topology optimization based graph convolutional network. In Proceedings of the Twenty-Eighth International Joint Conference on Artificial Intelligence, IJCAI. 2019, 4054-4061.

---

> > ### Author Response · Authors · 2019-11-15
> > **Reply to Official Review #3 Part 2**
> >
> > Q5. Concerning the Section on empirical analysis, it might be of interest to investigate whether with a proper number of layers a GCN would emulate a geom-GCN.
> >
> > Response:
> >
> > Thanks for the great suggestion. In this paper, we use a two-layer Geom-GCN currently for fair comparisons with GCN and GAT which both adopt a two-layer architecture. The performance of multi-layer GCNs is actually limited, which has been studied in JK-Net [Xu 2018]. Thus, it's a promising direction to design multi-layer graph neural networks. We will try to design deep Geom-GCNs, especially for disassortative graphs, in future work.
> >
> > [Xu 2018] Xu, K., Li, C., Tian, Y., Sonobe, T., Kawarabayashi, K. I., & Jegelka, S. Representation learning on graphs with jumping knowledge networks. International Conference on Machine Learning (ICML). 2018, 5449–5458.
> >
> > We thank the reviewer once again. And we have released anonymously the code of Geom-GCN, and the particular architectures of Geom-GCN are also reported in the revision. One can access the code via link: https://github.com/anonymous-conference-submission/geom-gcn/.

---

### Official Review · AnonReviewer2 · 2019-10-30
**Official Blind Review #2**

**Rating:** 6

**Review:**

GEOM-GCN: GEOMETRIC GRAPH CONVOLUTIONAL NETWORKS

The paper introduces a novel GCN framework, whose purpose is to overcome weaknesses of existing GCN approaches, namely loss of structural neighbor information and failure to capture important dependencies between distant nodes. The paper uses a mapping from nodes to an embedded space and introduces a second type of a neighborhood: a proximity in the embedded space. In the embedded space, a set of relations of nodes is defined. For each node v, the paper uses a 2-stage convolution scheme: 1) for each neighborhood type, the nodes in the same relation with v are combined; 2) the resulting nodes are again combined into a new feature vector. This approach allows one to overcome the issues described above. The experiments show that in most cases the approach outperforms the existing GCN solutions, sometimes with a large gap.

I have the following concerns about the paper:
-- My main concern is the learning time, which is an issue for a straightforward GCN implementation. There were multiple attempts to decrease it (GraphSAGE, FastGCN, etc.). Therefore, I would like to see running times on the presented graphs as well as on relatively large graphs (see e.g. https://arxiv.org/pdf/1902.07153.pdf for candidates). If some techniques were used to make the implementation faster, I would be good to include them in the paper (or, if they are standard, they should be referenced). At the very least, I believe it should be prioritized as a future direction.
-- It’s unclear why we should use the same latent space and the same τ for both N_g and N_s. I would expect that mapping into different spaces could provide better results: the two neighborhood types seem very different, and I don’t see why the neighbors should be aggregated in the same way. If using different spaces doesn’t provide an improvement, an explanation for this would be very useful.
-- α and β are defined and shown in Table 2, but they are never used (as it stands now, α and β can simply be removed). If the results in Table 3 correlate with them, then this dependence should be highlighted. In such case, it would also be better to move α and β to Table 3.
-- The paper uses 3 different node embedding strategies. These strategies can be combined in q with different weights (which can be learned as hyperparameters). Will it produce the best of 3 (or better) result?
-- “We use an embedding space of dimension 2 for ease of explanation” But what τ is used in the real implementation?
-- There are various GCN implementations; however, the comparison is performed with only 2 of them. I would like to see either comparison with more implementations, or the explanation why the comparison with the given two suffices.
-- Is it possible to make the implementation available?

While there are a lot of possible improvements, I believe that some of them can be addressed in a future research, and the paper’s novel approach is noteworthy in itself. My current verdict is 5/10, and I’ll be happy to improve it if the above issues are fixed.

Presentation issues:
-- The notation used in definition of m_v^l is unclear.
-- Why τ is a part of each node’s structural neighborhood? It’s a global function, isn’t it?
-- Introduction: I believe that the exact problems which GCNs solve (e.g. node classification) should be mentioned.
-- The flow in Section 2.1 is a bit weird. Namely, it says “To overcome the first weakness”, but the first wickness wasn’t stated in the previous paragraph (of course, one can deduce it, and it also was defined long ago, but it’s disturbing for a reader).
-- Figure 1B is confusing: it looks like the nodes from N_g(v) lie in a small region around v.
-- I think that splitting Figure 1C into 2 figures would make it clearer.


**Experience Assessment:**

I have read many papers in this area.

**Review Assessment: Checking Correctness Of Derivations And Theory:**

I assessed the sensibility of the derivations and theory.

**Review Assessment: Checking Correctness Of Experiments:**

I assessed the sensibility of the experiments.

**Review Assessment: Thoroughness In Paper Reading:**

I read the paper at least twice and used my best judgement in assessing the paper.

---

> ### Author Response · Authors · 2019-11-15
> **Reply to Official Review #2 Part 1**
>
> We thank the reviewer for the helpful comments. We have revised the paper according to the suggestions and would like to answer the reviewer’s questions as follows:
>
> Q1. My main concern is the learning time, which is an issue for a straightforward GCN implementation. There were multiple attempts to decrease it (GraphSAGE, FastGCN, etc.). Therefore, I would like to see running times on the presented graphs as well as on relatively large graphs (see e.g. https://arxiv.org/pdf/1902.07153.pdf for candidates). If some techniques were used to make the implementation faster, I would be good to include them in the paper (or, if they are standard, they should be referenced). At the very least, I believe it should be prioritized as a future direction.
>
> Response:
>
> It's a great suggestion. According to it, we add a new section (Section 4.3.3) in the revision to systematically analyze the running time of the proposed Geom-GCN. In this section, we firstly present the theoretical time complexity of Geom-GCN and then compare the real running time of GCN, GAT, and Geom-GCN. To decrease the running time, we think it's promising future work to apply the accelerating technologies for GCN (e.g., FastGCN and SCG) to Geom-GCN.
>
>
> Q2. It’s unclear why we should use the same latent space and the same $\tau$ for both N_g and N_s. I would expect that mapping into different spaces could provide better results: the two neighborhood types seem very different, and I don’t see why the neighbors should be aggregated in the same way. If using different spaces doesn’t provide an improvement, an explanation for this would be very useful.
>
> Response:
>
> Thanks for the suggestion. We add a new section (Section 4.3.2) in the revision to study how the embedded latent spaces influence the structural neighborhood.  To this end, we construct several new Geom-GCN variants, which use a combination of neighborhoods defined by different embedded spaces. From Table 5, we can observe that several variants achieve better performance than the original Geom-GCN with neighborhoods defined by only one embedded space. On the other hand, there are also many variants that have bad performances.  That is, the efficacy of Geom-GCN would depend on the selected embedded spaces. Thus, we think it's important future work to design an end-to-end framework that is able to automatically determine the right embedded spaces for Geom-GCN. We have finished the analysis on six small datasets in the current revision, and comprehensive results will be added in the future revision.
>
>
> Q3. $\alpha$ and $\beta$ are defined and shown in Table 2, but they are never used (as it stands now, $\alpha$ and $beta$ can simply be removed). If the results in Table 3 correlate with them, then this dependence should be highlighted. In such a case, it would also be better to move $\alpha$ and $\beta$ to Table 3.
>
> Response:
>
> Thanks for the suggestion. In the revision, we analysis the correlate between $\beta$ and the results in Table 4. We find that the latent space neighborhoods have larger contributions in disassortative graphs (with a small $\beta$) than assortative ones, which implies relevant information from disconnected nodes is aggregated effectively in the neighborhood of the latent space. Please see Section 4.3.1 in revision for details. And we removed $\alpha$ in the revision.
>
>
> Q4. The paper uses 3 different node embedding strategies. These strategies can be combined in q with different weights (which can be learned as hyperparameters). Will it produce the best of 3 (or better) result?
>
> Response:
>
> It's a very interesting idea. To implement the idea, we construct a new Geom-GCN variant that contains six neighborhoods defined by all the three embedded spaces. Then we evaluate the variant on the graph datasets. However, its performance is not good. We think it is because of two reasons, i) the variant is very hard to train since it has too many parameters; ii) there is too much irrelevant information from the six neighborhoods, which implies that the relevant information may be “washed out” by too much irrelevant information. We also release the code of this experiment in GitHub anonymously, please find the link in the following.
>
> Q5. “We use an embedding space of dimension 2 for ease of explanation” But what $\tau$ is used in the real implementation?
>
> Response:
>
> In the real implementation, we use the $\tau$ that we defined in Table 1 (section 3). In the revision, we exactly mention which $\tau$ is used in the experiment section.

---

> > ### Author Response · Authors · 2019-11-15
> > **Reply to Official Review #2 Part 2**
> >
> > Q6. There are various GCN implementations; however, the comparison is performed with only 2 of them. I would like to see either comparison with more implementations, or the explanation why the comparison with the given two suffices.
> >
> > Response:
> >
> > We only select the GCN and GAT as competitors because of two reasons. i) The goal of this paper is to address the two fundamental weaknesses of MPNNs (message passing neural networks), i.e., losing the structural information of nodes in neighborhoods and lacking the ability to capture long-range dependencies. From the perspective of the weaknesses, GCN and GAT are representative of existing MPNNs because all of them have the two weaknesses. ii) Although various GCN implementations have been proposed recently, GCN and GAT are two popular and stable benchmarks whose performance always very near the state-of-the-art.
> >
> >
> > Q7. Is it possible to make the implementation available?
> >
> > Response:
> >
> > We have released the code of Geom-GCN on GitHup anonymously. The Python implementation is based on DGL (Deep Graph Library) package. Please access the code via link: https://github.com/anonymous-conference-submission/geom-gcn/.
> >
> >
> > Q8. Presentation issues.
> >
> > Response:
> >
> > Thanks for the detailed suggestions. All the presentation issues have been modified in the revision carefully. Please see the details in the following.
> >
> >   Q8.1: The notation used in the definition of $m_v^l$ is unclear.
> >   R8.1: We update the description of $m_v^l$ to clarify it in Section 2C.
> >
> >   Q8.2: Why $\tau$ is a part of each node’s structural neighborhood? It’s a global function, isn’t it?
> >   R8.2: Yes, $\tau$ is a global function. We add a description for $\tau$ in Section 2B.
> >
> >   Q8.3: Introduction: I believe that the exact problems which GCNs solve (e.g. node classification) should be mentioned.
> >   R8.3: We describe the exact graph learning task, node classification, in the introduction of the revision.
> >
> >   Q8.4: The flow in Section 2.1 is a bit weird. Namely, it says “To overcome the first weakness”, but the first weakness wasn’t stated in the previous paragraph (of course, one can deduce it, and it also was defined long ago, but it’s disturbing for a reader).
> >   R8.4: We exactly describe the two weaknesses of MPNNs in Section 2.1 of the revision.
> >
> >   Q8.5: Figure 1B is confusing: it looks like the nodes from N_g(v) lie in a small region around v.
> >   R8.5: We add an explanation for the graph neighborhood in the caption of Figure 1 to clarify this confusion.
> >
> >   Q8.6: I think that splitting Figure 1C into 2 figures would make it clearer.
> >   R8.6: We modify the organization and caption of Figure 1 to make it clearer to a reader in the revision.
> >
> > We thank the reviewer once again.

---

### Public Comment · ~Yilun_Jin1 · 2019-09-30
**Novel idea but somehow unclear description of experiments**

I think the motivation is definitely novel in that it extends the conventional definition of neighbourhoods in graphs and hence addressed the problems of structural information preservation in GCN.

However, I do find the experiments are not well described. For example,
1. I suppose you use the semi-supervised setting as in GCN (Kipf and Welling 2017) and GAT (Velicikovic 2017), but you did not mention it in the paper, is that true?
2. You did not mention exact GCN architecture, i.e. hidden unit, output size, etc, other than a grid search, which makes it somehow hard to reproduce results.

Nonetheless I think it is an interesting and technically inspiring paper on GCN.

---

> ### Author Response · Authors · 2019-10-03
> **Thanks a lot for your comments!**
>
> Thanks a lot for your comments and affirmations.
>
> For your concerns about our experiments,
> Q1: "I suppose you use the semi-supervised setting as in GCN and GAT."
> R1: Yes. We compare the proposed Geom-GCN on transductive learning tasks, thus we use GCN and GAT with the semi-supervised setting. We will clarify it in the new version.
>
> Q2: "You did not mention exact GCN architecture, i.e. hidden unit, output size, etc, other than a grid search, which makes it somehow hard to reproduce results."
> R1: We will report the exact Geom-GCN architecture on each dataset in the new version. We conduct a fair parameter search for every method (i.e., GCN, GAT, Geom-GCN) on every dataset, thus their architecture is different on each dataset.  We will release the code of Geom-GCN after acceptance.

---

### Public Comment · ~Henry_Blackmore1 · 2019-10-20
**Nice work but some detailed results are necessary**

Nice work!
I have some questions about the details and model.
1. If the high-level aggregation function q is concatenation, the dimension of $m^{l+1}_v$ is not fixed, which may change with the number of virtual nodes. So how can you solve this? To my understanding, the input of q is a set of e, while the output should be fixed dimension. Hope you can explain about it.
2. The description of bi-level aggegration seems great, but it's better to provide experimental results on each block. I mean how is the performance of your model without high-level aggegration? just do simple aggegration in low-levlel to each node.
3. As for me, Geom-GCN has a good use of long-range information, which is not applicable for most existing works. well, i think a main challenge of making GCN deep is the use of long-range information, because former works with deep layers will absorb the same information for each node. What about Geom-GCN? I think with more layers like 5~6 or more, Geom-GCN can achiever better performance. Did you test this or can you provide results with deep layers, making this paper better?

---

> ### Author Response · Authors · 2019-10-21
> **Thanks a lot for your helpful comments!**
>
> Thanks a lot for your helpful comments!
>
> I try my best to elaborate on your concerns.
> Q1: Is the size of $m_v^{l+1}$, the output of high-level aggregation function q, fixed?
> R1: Yes, the size is fixed. Your concern originates from missing virtual nodes because there may be no node in certain blocks in the neighborhood. And the changing of the number of virtual nodes leads to the different size of input of q, which is a set of features of virtual nodes. To address this issue, we specify a zero-vector as the feature of the missing virtual node in our model, thereby keeping the same size of input/output of q.
>
> Q2: To provide experimental results with only low-levlel aggregation in each block
> R2: It's a very good suggestion. By doing so, we can analyze the contribution of each block for the final task, e.g., node classification.
>
> Q3: Did you test Geom-GCN with deep layers (e.g., 5~6 or more) and provide the results to make this paper better?
> R2: It's a very good suggestion. We did not test Geom-GCN with deep layers. We use a two-layer Geom-GCN for fair comparisons with GCN and GAT which both adopt a two-layer architecture. We will design deep Geom-GCN and test it, especially on disassortative graphs, in future work.
>
> The paper will be modified in the future version according to your comments. Thank you very much!

---

### Public Comment · ~Jiong_Zhu1 · 2019-12-22
**Bug in the Published Code Repository for Parsing the Actor Dataset**

We think there is a bug in the published code which will lead to incorrect parsing of the Actor dataset (in the code it is referred as "film"), and as a result, the performance reported for the Actor dataset in Table 3 of the paper is likely to be incorrect if the experiments are based on the published code.

In file "utils_data.py" in the repo, line 61, at commit bd0acd9:

feature_blank[np.array(line[1].split(','), dtype=np.uint8)] = 1

The "dtype" here should not be "np.uint8" since this array is supposed to contain values larger than 255. From our understanding, since the dimension of feature vectors in this dataset is around 931, the maximum possible value of this array should be around 930. Thus, if "np.uint8" is used as data type, an integer overflow problem will occur and the feature vectors will be incorrectly loaded; it should be fixed to a data type like "np.uint16".

The size of the array allocated in line 60 may also need to be fixed accordingly.

---

> ### Author Response · Authors · 2019-12-28
> **Thank You for the Bug Report! Increased Lead Over Baseline Methods on the Actor Dataset (Table 3)**
>
> Dear Mr. Zhu,
>
> Thank you very much for your detailed report regarding a bug in the published code for Geom-GCN. We were able to reproduce the overflow in utils_data.py and have fixed the published code accordingly. Please see https://github.com/anonymous-conference-submission/geom-gcn/ for the most up-to-date version of the published code for Geom-GCN.
>
> We have also rerun all affected experimental trials and have found that the performance advantage of Geom-GCN over baseline methods have increased. In particular, the updated results for GCN, GAT, and Geom-GCN on the Actor dataset (Table 3) are as follows:
>
>               Mean Classification Accuracy (Percent)
>
>      Model                  Old   ->  New          Absolute Change
>
> GCN                         27.92 -> 26.86                -1.06%
> GAT                          28.15 -> 28.45                +0.30%
>
> GeomGCN-I           28.65 -> 29.09                +0.44%
> GeomGCN-P          31.28 -> 31.63                +0.35%
> GeomGCN-S          29.83 -> 30.30                +0.47%
>
> Sincerely,
> Geom-GCN Authors

---

### Public Comment · ~Deyu_Bo1 · 2020-04-23
**The error results of Table 3**

In the source code of 'utils_data.py', line 96, the adjacency matrix is not symmetrical, which is unfair for GCN and GAT.
When using a symmetric adjacency matrix, the performances of GCN and GAT are greatly improved.

---

### Public Comment · ~Zhitao_Ying1 · 2020-04-30
**The experiment is not in semi-supervised setting as in the previous author response**

There was a question of whether  the semi-supervised setting of GCN and GAT are used for experiments. The authors' answer is contradictory to the description of the paper.

The paper actually uses *Random* split (with 60%, 20%, 20% ratio), rather than the split for semi-supervised setting in GCN (fixed small set of training nodes).

Would appreciate if the authors also publish the settings and results in the standard semi-supervised setting, to support the claim and make the results reproducible in the standard setting for Cora, CiteSeer etc. as such split is done for the previous papers.

---

### Public Comment · ~Zhengyang_Wang1 · 2020-06-01
**MLP outperforms all GNNs on four of the disassortative datasets**

We find that a simple 2-layer MLP outperforms all GNNs on four of the disassortative datasets as shown below. More details are provided in our paper (https://arxiv.org/pdf/2005.14612.pdf).

Datasets            Actor           Cornell           Texas           Wisconsin
GCN                   30.3               54.2                61.1                 59.6
Geom-GCN       31.6               60.8                67.6                 64.1
Geom-GCN-s    34.6               75.4                73.5                 80.4
MLP                    35.1               81.6                81.3                 84.9

---

### Decision · Program_Chairs · 2019-12-19

**Decision:**

Accept (Spotlight)

**Comment:**

This paper is consistently supported by all three reviewers and thus an accept is recommended.